# Genome-Wide Identification of Members of the Soybean *CBL* Gene Family and Characterization of the Functional Role of *GmCBL1* in Responses to Saline and Alkaline Stress

**DOI:** 10.3390/plants13101304

**Published:** 2024-05-09

**Authors:** Feng Jiao, Dongdong Zhang, Yang Chen, Jinhua Wu

**Affiliations:** College of Agriculture, Heilongjiang Bayi Agricultural University, Daqing 163319, China; jiaofeng_1980@126.com (F.J.); a230224127@126.com (D.Z.); lbt22222@163.com (Y.C.)

**Keywords:** soybean (*Glycine max* L.), CBL, salt stress, genome-wide analysis, expression patterns, transgenic plants

## Abstract

Calcium ions function as key messengers in the context of intracellular signal transduction. The ability of plants to respond to biotic and abiotic stressors is highly dependent on the calcineurin B-like protein (CBL) and CBL-interacting protein kinase (CIPK) signaling network. Here, a comprehensive effort was made to identify all members of the soybean *CBL* gene family, leading to the identification of 15 total genes distributed randomly across nine chromosomes, including 13 segmental duplicates. All the *GmCBL* gene subfamilies presented with similar gene structures and conserved motifs. Analyses of the expression of these genes in different tissues revealed that the majority of these *GmCBLs* were predominantly expressed in the roots. Significant *GmCBL* expression and activity increases were also observed in response to a range of stress-related treatments, including salt stress, alkaline stress, osmotic stress, or exposure to salicylic acid, brassinosteroids, or abscisic acid. Striking increases in *GmCBL1* expression were observed in response to alkaline and salt stress. Subsequent analyses revealed that *GmCBL1* was capable of enhancing soybean salt and alkali tolerance through the regulation of redox reactions. These results offer new insight into the complex mechanisms through which the soybean *CBL* gene family regulates the responses of these plants to environmental stressors, highlighting promising targets for efforts aimed at enhancing soybean stress tolerance.

## 1. Introduction

Soybeans are a key leguminous crop, serving as a major source of edible protein and oil for consumption by humans and use in the production of animal feed throughout the world [1]. Rising levels of soil salinity, however, are adversely impacting global soybean yields [2], threatening soybean cultivation. The enhancement of soybean tolerance for saline–alkaline conditions is vital as a means of improving agricultural productivity on salinized lands.

Salt stress can result from both osmotic disturbances and the ionic toxicity posed by sodium ions (Na^+^) [3]. Cytoplasmic calcium ion (Ca^2+^) concentrations rise immediately in response to elevated levels of salinity [4,5]. Calcium functions as a major mediator of signal transduction in plants, governing their growth, development, and ability to respond appropriately to environmental stressors [6,7]. This burst of Ca^2+^ signaling activity has been demonstrated to be triggered by the binding of extracellular sodium ions to cell-surface-exposed glycosyl inositol phosphoryl ceramides (GIPCs) [8]. Research to date has led to the characterization of four prominent Ca^2+^ sensors, including calcineurin B-like protein (CBL), calcium-dependent protein kinase (CDPK), calmodulin (CAM), and calmodulin-like protein (CML) [9]. With the exception of the kinase-domain-containing CDPKs, these Ca^2+^ sensors lack any enzymatic domains, suggesting that they serve as intermediaries in the process of cellular signal transmission to targets downstream [10]. The CBL protein was first identified in *Arabidopsis thaliana*, and it shares a high degree of similarity with the regulatory subunit of the mammalian calcineurin protein [11,12]. CBLs are composed of a structural domain consisting of four conserved helix–loop–helix motifs (EF−hands) that serve as a site for calcium binding, allowing for the capture of Ca^2+^ and for interaction with CIPKs (CBL-interacting protein kinases) [13,14]. Canonical EF−hand motifs are generally composed of Asp, Asp, Asp, Thr, Asp, and Glu in the ligand coordinating positions [15]. A serine present within the conserved PFPF motif of CBL has been established as a site of CIPK-mediated CBL phosphorylation [16]. Some CBL proteins contain a site that undergoes N-myristoylation, which is required for the functional role that these plants play in the enhancement of tolerance to salt stress [17].

Together, CBL and CIPK proteins comprise a calcium signaling network that is central to the ability of plants to respond appropriately to abiotic stressors [18,19]. Ten CBLs that interact with twenty-six CIPKs have been identified in *A. thaliana* to date [20,21]. The first CBL-CIPK network to have been characterized is the salt overly sensitive (SOS) pathway, which entails signaling mediated by a complex of CBL4 (SOS3) and CIPK24 (SOS2). Interactions between SOS3 and SOS2 result in the phosphorylation of SOS1, a Na/H antiporter located at the plasma membrane, activating it and thereby enhancing the export of Na^+^ [22,23,24]. Research focused on the *AtCBL1* gene in *A. thaliana* has revealed that it is inducible in response to stressors including salt, cold, or drought stress conditions. *AtCBL1* overexpression led to the establishment of plants that were better able to resist drought and salt stress, although they were more sensitive to freezing conditions, thereby establishing *AtCBL1* as a positive regulator of drought and salt responses, whereas it negatively regulates cold responses [25]. *AtCBL1* overexpression can protect against water loss in *A. thaliana* while promoting the upregulation of a series of early stress-response-related transcription factors and genes important for adapting to stress conditions, thus regulating DREB-type-related transcription factors [26]. *AtCBL5* has similarly been found to positively regulate drought and salt stress in plants such that they can better contend with these stressors when it is overexpressed [27]. *A. thaliana* plants overexpressing *ZmCBL9* from *Zea mays* L. were better able to tolerate osmotic and salt stress. At the molecular level, ZmCBL9 was found to interact with several ZmCIPK proteins, supporting its importance as an abiotic stress response regulator [28]. *CBL10* has also been reported to be central to the regulation of stress-related signal transduction in several model plant species, activating appropriate Ca^2+^ regulatory channels in response to particular stressors. CBL10 can also regulate respiratory burst oxidase homolog protein activity, thereby modulating rapid long-distance signal transmission [29]. The evidence published to date thus highlights the status of CBL proteins as essential regulators of abiotic stress responses in plants, emphasizing their importance in an array of stress-related signaling pathways.

Plants engage in myriad metabolic and growth-related processes over the course of their life cycle, and CBL proteins help appropriately govern and direct the related responses. Ca^2+^ ions are vital second messengers that control activities including the germination of pollen and tube growth, both of which entail the expression of high levels of *AtCBL1* and *AtCBL9* expression in *A. thaliana*. When either of these genes are overexpressed, this reportedly suppresses the rates of pollen germination and tube elongation, particularly when potassium concentrations are elevated [30]. Studies of the apple gene *MdCBL5* have indicated that it is strongly expressed in pollen, controlling the Ca^2+^ ion gradient at the pollen tube apex and thereby governing tube growth activity [31]. CBL2 and CBL3, which localize the vacuolar membrane, can also reportedly influence *A. thaliana* nutritional growth and reproductive development, with the effect being most pronounced when both of the associate genes are mutated [32]. Indeed, there is strong evidence for the importance of *CBL2* and *CBL3* in the context of seed development such that embryonic development is markedly disrupted when both are completely absent [33]. Members of the CBL protein family are thus indispensable regulators of plant development.

Despite intensive research characterizing the *CBL* gene family in model plant species including *A. thaliana [21]*, rice (*Oryza sativa*) [21], canola (*Brassica napus*) [34], and grapevine (*Vitis vinifera*) [35], there have not been any corresponding genome-wide analyses of this gene family in soybean (*Glycine max* var. Williams 82), even though its genome has been sequenced [36]. The present study was thus developed to address this knowledge gap by assessing the gene structures, expression patterns, evolution, and syntenic relationships of members of the soybean *CBL* gene family, potentially as pertaining to their biological roles in the context of saline–alkaline stress. To that end, the responses of *CBL* genes to factors including abiotic stress conditions and hormone treatment were systematically analyzed, with supporting analyses of conserved motifs, collinearity, and patterns of gene expression being conducted. Soybean plants expressing elevated levels of *GmCBL1* were also established, and their responses to saline–alkaline stress were evaluated. The overall goal of this study was to determine the regulatory functions that govern the activity of soybean *CBL* genes and the importance of members of this family in stress adaptation, thereby informing efforts aimed at establishing soybean crops with greater stress resistance.

## 2. Results

### 2.1. Identification of the G. max CBL Gene Family

The full-length protein sequences and conserved domains of 10 *A. thaliana* CBL proteins were selected for use as BLAST queries against the Phyozome database, leading to the identification of 15 putative soybean *CBL* genes designated as *GmCBL1*-*15* (Table 1). Among the several analyzed plant species, the genome of *T. aestivum* encoded the highest number of *CBL* genes, whereas *G. max* encoded the highest number of *CBL* genes among the legume family members (Appendix A). High homology was observed between the *G. max CBL* gene and *G. soja CBL* gene (Appendix A). *GmCBL1*–*GmCBL15* ranged from 639 to 798 base pairs in terms of coding sequence length, with predicted protein lengths ranging from 213 to 266 amino acids. These *GmCBLs* exhibited isoelectric points from 4.31 to 4.76 and molecular weights between 24.4 and 30.65 kDa (Table 1). None of these *GmCBLs* presented with notably distinct sequence lengths or molecular weight values as compared to other members of this family. Cell-PLoc-based predictions suggested that the subcellular localization of the proteins encoded by these genes varied substantially, with 5, 1, 1, and 8, respectively, localizing primarily to the nucleus, extracellular matrix, chloroplast, and cytoplasm (Table 1).

### 2.2. Phylogenetic Analyses and Classification of CBL Genes

To explore the classification of and the evolutionary relationships among the *GmCBLs,* the full-length protein sequences of these *GmCBLs* were aligned to homologous *CBLs* encoded by *S. bicolor* (*SbCBL1*-*8*), *A. thaliana* (*AtCBL1-10*), *O. sativa* (*OsCBL1*-*10*), *Z. mays* (*ZmCBL1*-*9*), *P. vulgaris* (*PvCBL1*-*10*), *V. unguiculata* (*VuCBL1*-*8*), *T. aestivum* (*TaCBL1*-*19*), and *G. soja* (*GsCBL1*-*14*), enabling the construction of a phylogenetic tree (Figure 1A). These plant *CBLs* were classified into nine distinct clades (I–IX), with the *GmCBLs* being unevenly distributed across these clades. The greatest number were located in clade I, while two *GmCBLs* each were located in clades IV and VI, and none were located in clades V, VII, VIII, or IX. These patterns of *GmCBL* distribution emphasize the diversity of this *GmCBL* gene family. Strikingly, the identified *GmCBL* isoforms were found to be closely related to homologous genes encoded by *G. soja* within each of these clusters, consistent with the fact that these species are both members of the same leguminous family (Figure 1A).

### 2.3. GmCBL Genes Syntenic Relationship, Gene Structure, and Protein Structure Analyses

To gain further insight into the structural characteristics of members of the *GmCBL* family, the exon–intron patterns and sizes of these genes were next evaluated. Those *GmCBL* genes in a given cluster presented with a high degree of similarity with respect to their exon–intron structures (Figure 1B), particularly with respect to the exon numbers. For example, the *GmCBLs* in cluster I all presented with eight exons, and only minimal differences in exon lengths were present among the members of this cluster. Exon numbers were similarly conserved among the *GmCBLs* in clusters II, III, and VI (n = 8), emphasizing the highly conserved nature of these genes in terms of their sequences and exon–intron organization within each of these phylogenetic groups. To better understand the evolution of this gene family, potential duplication events within the *G. max* genome were analyzed (Figure 1C), leading to the identification of 13 such duplication events involving 14 *GmCBL* genes (*GmCBL1*, *GmCBL2*, *GmCBL3*, *GmCBL4*, *GmCBL5*, *GmCBL6*, *GmCBL7*, *GmCBL8*, *GmCBL9*, *GmCBL10*, *GmCBL11*, *GmCBL12*, *GmCBL13*, and *GmCBL15*). Duplications thus appear to have played a key role in *GmCBL* gene family expansion. These 15 GmCBL family proteins were also found to harbor four conserved domains, including the EF−hand_1, EF−hand_5, EF−hand_7, and EF−hand_8 domains (Figure 1D). In addition to contributing to the overall structural diversity of these GmCBLs, these domains may play vital roles in the functional differentiation of members of this family.

### 2.4. GmCBL Chromosomal Distribution and Motif Analyses

These 15 *GmCBL* genes were found to be randomly distributed across chromosomes (Chr) throughout the *G. max* genome (Appendix A). Specifically, *GmCBL1*, *GmCBL4*, *GmCBL10*, *GmCBL11*, and *GmCBL15* were, respectively, encoded on Chr 4, 6, 9, 11, and 18, while *GmCBL2* and *GmCBL3* were encoded on Chr 5; *GmCBL5* and *GmCBL6* on Chr 7; *GmCBL7*, *GmCBL8*, and *GmCBL9* on Chr 8; and *GmCBL12*, *GmCBL13*, and *GmCBL14* on Chr 17. These genes exhibited some tendency towards clustering, such that they were predominantly located at the tops or bottoms of individual chromosomes. When the MEME tool was used to conduct a motif analysis of conserved motifs across these GmCBL proteins (Appendix A and Figure 2A), 10 conserved motifs ranging from 8 to 50 amino acids in length (motifs 1–10) were identified in various combinations across this GmCBL protein family. The highest numbers of conserved motifs were evident in GmCBL5 and GmCBL8. The majority of the GmCBLs included motifs 1, 2, 3, 4, and 6. Moreover, GmCBLs clustered within the same phylogenetic branch exhibited consistent motif compositions, highlighting the evolutionary and functional similarity within these clusters. The 3D structures of these GmCBL EF−hand_7 domains were composed of three α-helices (α1, α2, and α3) and a β-sheet1 that was roughly parallel to β-sheet2 (Figure 2B). Together, the results of this motif analysis offer new insight into the conserved structural elements that contribute to *GmCBL* gene family functional diversity.

### 2.5. Cis-Acting Element Analyses of GmCBL Gene Promoters

To identify potential *cis*-acting elements involved in the regulation of *GmCBL* gene transcriptional activity, a 2.0 kb promoter region upstream of the ATG start codon for each of these *GmCBLs* was analyzed. This approach led to the identification of stress-responsive *cis*-acting elements upstream of all genes other than *GmCBL6* and *GmCBL12* (Figure 3). The drought-responsive MBS element was detected upstream of *GmCBL7*, *GmCBL8*, *GmCBL14*, and *GmCBL15*. The anoxia-inducible ARE element was upstream of *GmCBL1*, *GmCBL2*, *GmCBL3*, *GmCBL4*, *GmCBL7*, *GmCBL8*, *GmCBL10*, *GmCBL11*, *GmCBL13*, and *GmCBL15*. TC-rich repeats involved in defense and stress responses were upstream of *GmCBL1*, *GmCBL2*, *GmCBL4*, *GmCBL5*, *GmCBL9*, *GmCBL11*, and *GmCBL15*. A low-temperature-responsive (LTR) element was identified as being upstream of *GmCBL3* and *GmCBL15*. Meristem-expression-related CAT and NON elements were detected upstream of *GmCBL1*, *GmCBL3*, *GmCBL5*, *GmCBL11*, and *GmCBL14*. These detailed bioinformatics results emphasize the key roles that these *GmCBLs* likely play as mediators of plant growth and stress responses, while suggesting that the expression of these genes is carefully regulated in response to particular environmental cues.

### 2.6. Multiple Sequence Alignment

As a means of probing the conserved regions present within the proteins encoded by these 15 *GmCBL* genes, multiple sequence alignment of these *G. max* protein sequences was next performed. Most of these GmCBLs were found to harbor the EF−hand_7 domain, with the exception of GmCBL7 and GmCBL8 (Figure 4A). GmCBL7 was found to harbor the EF−hand_5 domain, while GmCBL8 and GmCBL10 both harbored the EF−hand_5 and EF−hand_7 domains. The EF−hand_8 domain was present in nine of these GmCBLs and was only absent from GmCBL6, GmCBL7, GmCBL11, GmCBL12, GmCBL13, and GmCBL14. The EF−hand_1 domain was unique to GmCBL14, while C-terminal FPSF motifs were present in 11 of these GmCBLs. Notably, the N-terminal MGCXXSK/T motif, which has been established as the myristoylation domain a specialized CBL protein domain [37], was present in GmCBL1, GmCBL2, GmCBL4, GmCBL11, and GmCBL13. These protein sequence analyses offer new insight into the conserved motifs and domains present within these soybean CBL proteins, providing additional information regarding their structural diversity and potential functional roles.

### 2.7. Tissue-Specific Expression Profiling of GmCBLs

Next, the patterns of *GmCBL* expression across different tissues were analyzed using high-throughput sequencing data from the Phytozome database, analyzing the expression of these genes in roots, stems, leaves, pods, seeds, and flowers (Figure 4B). These genes were expressed in distinct patterns, with higher relative *GmCBL11* and *GmCBL3* mRNA levels in flowers, while *GmCBL6*, *GmCBL2*, *GmCBL1*, and *GmCBL5* were expressed more robustly in roots, *GmCBL10* was highly expressed in seeds, and *GmCBL13* was highly expressed in pods. Relative to other members of this gene family, *GmCBL13* was expressed at lower levels across these analyzed tissues. Based on their expression patterns across tissues, these *GmCBLs* were classified into five major expression groups (I–V) (Figure 4B), with this categorization largely aligning with the phylogenetic clades established above. Similar patterns of *GmCBL* expression were observed within each of these subgroups. Overall, the tissue-specific *GmCBL* expression patterns emphasize the varied roles that these genes play in the growth and development of soybean plants.

### 2.8. CBL Gene Synteny Analysis

Collinear relationships among members of the *GmCBL* gene family and *CBL* genes encoded by other plants were examined by selecting a diverse range of representative plants including a dicotyledonous plant (*A. thaliana*), three monocotyledonous plants (*Z. mays*, *S. bicolor*, and *O. sativa*), and three leguminous plants (*P. vulgaris*, *G. soja*, and *V. unguiculata*). A total of 8, 7, 28, 9, 44, 18, and 31 collinear blocks were, respectively, identified in *Z. mays*, *O. sativa*, *P. vulgaris*, *S. bicolor*, *G. soja*, *A. thaliana*, and *V. unguiculata*. Seven collinear diagrams were established to examine the collinearity between *G. max* and these representative plants, leading to the identification of homologous gene pairs linked by color-coded lines (Figure 5 and Appendix A). Of the 15 *GmCBLs* identified in this study, 8 did not exhibit collinearity with *Z. mays*, *S. bicolor*, or *O. sativa*, suggesting that these homologous gene pairs may have emerged following the monocot–dicot divergence. Higher numbers of linear collinear genes were observed among leguminous plants, supporting their closer evolutionary relationships as compared to the other groups of plants included in this study.

### 2.9. Characterization of GmCBL Transcriptional Responses to Abiotic Stress and Hormone Stimulation

Research focused on several model plants has established the essential role that *CBL* genes play in the regulation of abiotic stress responses [18,38,39]. To test the ability of these *GmCBL* genes to respond to abiotic stress conditions, changes in the expression of these genes in response to a range of stress conditions, including salt stress (150 mM NaCl), alkaline stress (100 mM NaHCO_3_), drought stress (200 mM mannitol), and osmotic stress (20% PEG), were assessed (Figure 6). The transcriptional responses of these *GmCBLs* varied among the groups. Specifically, *GmCBL1*, *GmCBL3*, *GmCBL5*, *GmCBL6*, *GmCBL7*, *GmCBL8*, *GmCBL9*, *GmCBL10*, *GmCBL11*, *GmCBL13*, and *GmCBL14* were expressed at significantly higher levels under salt stress conditions as compared to control conditions, with this effect being most pronounced for *GmCBL1*. *GmCBL1*, *GmCBL3*, *GmCBL7*, *GmCBL8*, *GmCBL9*, *GmCBL11*, *GmCBL12*, *GmCBL13*, and *GmCBL14* were significantly upregulated under alkaline stress conditions relative to control conditions, with *GmCBL1* again exhibiting robust and rapid upregulation in response to alkaline conditions. Drought stress was associated with a significant upregulation of *GmCBL1*, *GmCBL4*, *GmCBL6*, GmCBL8, *GmCBL10*, *GmCBL11*, *GmCBL12*, and *GmCBL14* relative to control conditions, while *GmCBL2*, *GmCBL7*, *GmCBL10*, *GmCBL11*, *GmCBL12*, *GmCBL13*, *GmCBL14*, and *GmCBL15* were significantly upregulated under osmotic stress conditions. *CBL* gene expression patterns are thus likely associated with the abiotic stress responses engaged by soybean plants, with *GmCBL1* in particular holding promise as a potentially vital regulator of saline–alkaline stress responses.

Hormones function as essential chemical messengers in plants that govern diverse processes, including growth and abiotic stress responses [40]. As most of the identified *GmCBL* genes were predicted to be regulated by hormone-signaling-related *cis*-acting elements (ABRE, CGTCA, P-box, TATC, TCA, TGA, TGACG, CGTCA) (Figure 3), the transcriptional responses of these genes to hormone treatment (ABA, SA, JA, or BR at 100 μmol L^−1^) in soybean roots were next evaluated (Figure 7). These genes exhibited a higher degree of variability with respect to how they responded to these different hormone treatments. For instance, *GmCBL3*, *GmCBL7*, *GmCBL13*, *GmCBL10*, and *GmCBL14* exhibited a high degree of JA sensitivity, whereas *GmCBL1*, *GmCBL7*, *GmCBL10*, *GmCBL11*, and *GmCBL13* were highly responsive to ABA. Moreover, *GmCBL4*, *GmCBL5*, *GmCBL6*, *GmCBL9*, GmCBL10, *GmCBL11*, *GmCBL12*, *GmCBL14*, and *GmCBL15* presented with a high degree of BR sensitivity, whereas *GmCBL1*, *GmCBL2*, *GmCBL3*, *GmCBL7*, *GmCBL8*, *GmCBL13*, and *GmCBL11* were highly sensitive to SA. The changes in the expression of these different *GmCBL* genes to particular hormones support the predicted identification of hormone-responsive *cis*-acting elements within the promoter regions regulating their expression. The correlation analysis of 15 *GmCBL* genes under different hormone stress and abiotic stress conditions further showed that the highest correlation coefficient between 150 mM NaCl and 100 μmol ABA was 0.52. It is worth noting that there was a negative correlation between 150 mM NaCl and 100 μmol BR (Appendix A).

### 2.10. GmCBL1 Subcellular Localization and GmCBL1 Overexpression Enhances Soybean Saline–Alkaline Tolerance

Because of the strong and rapid response of the *GmCBL1* gene to saline–alkaline stress, the gene was selected for further experiment. The GFP fusion expression vector was constructed to identify the subcellular location of the protein encoded by *GmCBL1* and transformed into *A. thaliana* protoplasts (Figure 8A). Under a confocal microscope, the GFP signal of the control vector could be seen everywhere in the whole protoplast cells, while the *GmCBL1*-GFP fusion protein signal was located in the cytoplasm (Figure 8B), which was consistent with the predicted results (Table 1). To directly test the functional role of *GmCBL1* under conditions of saline–alkaline stress, *Agrobacterium rhizogenes* transformation was used to establish soybean hairy roots overexpressing *GmCBL1* (OHRs), with PCR confirming the identities of 10 positive transgenic plants (Appendix A). To determine how overexpressing *GmCBL1* impacts saline–alkaline tolerance in soybean seedlings, 3-week-old CHR and OHR plants were exposed to 0, 75 mM NaCl and 75 mM NaHCO_3_ for 5 days. The CHR seedlings presented with severe growth inhibition and shrinkage (Figure 9), whereas the *GmCBL1*-OHR plants exhibited markedly higher levels of saline–alkaline stress resistance, as demonstrated by their longer root length and greater root fresh and dry weight values (Figure 9B–D). To further probe the role that *GmCBL1* plays in redox processes, four antioxidant enzyme genes were analyzed, with qPCR analyses being used to probe their expression in *GmCBL1*-OHR plants under saline–alkaline stress conditions. In this experiment, the antioxidant genes *GmSOD1*, *GmAPX1*, *GmCAT1*, and *GmGSH1* were upregulated, as was the *GmCPA1* gene encoding a Na^+^/H^+^ antiporter.

## 3. Materials and Methods

### 3.1. Plant Materials and Treatment Conditions

The DN50 soybean cultivar from the Soybean Breeding Research Center of Northeast Agricultural University (Harbin, Heilongjiang, China) was cultivated in the experimental plantation of Northeast Agricultural University. The effects of abiotic stress conditions on *GmCBL* expression were assessed using seedlings at the second trifoliolate stage cultivated at 23 ± 2 °C with a 12 h photoperiod. The abiotic stress treatment entailed exposure to solutions of 20% PEG (molecular weight: 6000), 150 mM NaCl, 100 mM NaHCO_3_, and 200 mmol L^−1^ mannitol. At 0, 1, 3, 6, 12, and 24 h post treatment, roots were collected from these plants, analyzing untreated roots as a control. Analyses of *GmCBL* gene expression changes in response to hormone treatment were assessed via the treatment of plants with JA, SA, ABA, or BR at 100 μmol L^−1^, collecting roots at the same time points as for the abiotic stress treatment. The above experiments were carried out in the indoor greenhouse of Northeast Agricultural University (longitude: 126.727556, dimension: 45.744028).

### 3.2. Soybean CBL Gene Family Identification

A comprehensive analysis of soybean *CBL* genes was performed by conducting a BLASTP search against the Phytozome (https://phytozome-next.jgi.doe.gov/, accessed on 10 February 2023) and NCBI (http://www.ncbi.nlm.nih.gov, accessed on 10 February 2023) databases, using *CBL* sequences from *A. thaliana* as queries. The retrieved candidate soybean *CBLs* were genes with an E-value < 10^−10^ and a >90% sequence identity. The SMART (http://smart.embl.de/, accessed on 10 February 2023) and Pfam (https://www.ebi.ac.uk/interpro/, accessed on 10 February 2023) databases were used for additional validation, confirming that these proteins harbor the essential domains associated with CBL proteins, including PF00036, PF13499, PF13833, and PF13202. Genetic *GmCBL* characteristics, including coding length, chromosomal location, and protein length, were determined based on the Phytozome database (https://phytozome-next.jgi.doe.gov/, accessed on 20 February 2023). Molecular weights and isoelectric point values for these *GmCBLs* were determined with the SIB database (https://web.expasy.org/compute_pi/, accessed on 20 February 2023). Predictive CBL protein subcellular localization analyses were performed with the Cell-PLoc tool (http://www.csbio.sjtu.edu.cn/bioinf/Cell-PLoc-2/, accessed on 20 February 2023).

### 3.3. Evolutionary, Tertiary Structure, Gene Structure, Promoter, Conserved Domain, and Synteny Analyses of GmCBLs

To explore *GmCBL* evolution, the full-length amino acid sequences of CBL proteins from *G. max*, *Z. mays*, *A. thaliana*, *O. sativa*, *P. vulgaris*, *G. soja*, *T. aestivum*, *V. unguiculata*, and *S. bicolor* underwent ClustalW alignment [41]. MEGA 5.0 was then used to generate an unrooted neighbor-joining phylogenetic tree with 1000 bootstrap replicates [42]. A MEME analysis was used to identify conserved protein motifs in *GmCBLs* [43]. The Phytozome database was used to determine the genomic locations of *GmCBLs*, with TBtools being used for visualization [44]. Details pertaining to exon–intron structures were obtained from the *G. max* GFF file, while the gene structures of *GmCBLs* were assessed using TBtools [44]. *Cis*-acting elements within the 2.0 kb promoter region upstream of the ATG start codon for each of these *GmCBL*s were identified by downloading these promoter sequences from the Phytozome database (https://phytozome-next.jgi.doe.gov//, accessed on 22 February 2023) and then utilizing the PlantCARE database (http://bioinformatics.psb.ugent.be/webtools/plantcare/html//, accessed on 23 February 2023) to predictively identify these *cis*-acting elements [45,46]. Syntenic blocks among *CBL* genes from *G. max*, *Z. mays*, *A. thaliana*, *O. sativa*, *G. soja*, *V. unguiculata*, *P. vulgaris*, and *S. bicolor* were identified with TBtools [44]. WebLogo (https://weblogo.threeplusone.com//, accessed on 24 February 2023) was used to generate conserved motif logos. The AlphaFold Protein Structure Database (https://alphafold.ebi.ac.uk//, accessed on 4 March 2023) was used to evaluate the *GmCBL* tertiary structure. Gene IDs and other information pertaining to these *CBL*s are presented in Table 1 and Appendix A.

### 3.4. GmCBL Expression Analyses

Patterns of *GmCBL* expression in various tissues were analyzed by assessing high-throughput sequencing data available through the Phytozome database (https://phytozome-next.jgi.doe.gov/, accessed on 26 February 2023), with analyzed tissues including the roots, stems, leaves, pods, seeds, and flowers. Data were presented as a hierarchically clustered heatmap generated with TBtools [44], with all expression values having been subjected to log_2_ transformation and normalization.

### 3.5. qPCR

Collected samples were homogenized in liquid nitrogen, after which Trizol (Invitrogen, Carlsbad, CA, USA) was used to extract total RNA. Primers specific for the 15 *GmCBL* genes were generated with the Primer-BLAST tool in Primer 3 Plus (https://www.primer3plus.com/, accessed on 25 March 2023). Three biological replicates per tissue were independently analyzed, with *GmACTIN4* serving as a normalization control and the 2^−ΔΔCT^ method being used to quantify gene expression [47]. See Appendix A for further details regarding the utilized primers.

### 3.6. GmCBL1 Overexpression in Soybean Hairy Roots and Subcellular Localization Assays

Specific primers (Appendix A) were used to amplify the *GmCBL1* gene from the DN50 soybean cultivar, after which the amplified sequences were integrated into the GFP-tagged pSOY1 vector under the control of the *CaMV35S* promoter using a high-precision KOD-Plus-Neo polymerase (TOYOBO, Osaka, Japan) [48]. PCR and fluorescence microscopy were used to evaluate the transformation efficiency, selectively removing any non-transgenic hairy roots from seedlings. Transgenic lines exhibiting hairy roots of similar length were selected and treated for 5 days with NaCl (75 mM) or NaHCO_3_ (75 mM). The fresh root weight and root length were then quantified for these transgenic plants. Over 10 independent hairy root lines were analyzed to ensure these results were reliable, allowing for the comprehensive establishment of how overexpressing *GmCBL1* impacts soybean responses to salt stress. *A. thaliana* mesophyll protoplasts were transfected with *Agrobacterium tumefaciens EHA105* strains containing the *35S*::*GmCBL1*-GFP and *35S*::GFP constructs and subsequently subjected to subcellular localization analysis after a 3-day incubation period.

### 3.7. Statistical Analyses

At least three biological replicates were analyzed for all the data, which are presented as means ± standard deviation (SD). The relationship between plant hormone and abiotic stress was analyzed by R-studio (RStudio-2023.12.1) correlation analysis [49]. SPSS 22.0 was used to analyze all the data via Student’s *t*-tests.

## 4. Discussion

Through the phylogenetic tree construction of 103 CBL proteins from different plants (Figure 1 and Appendix A), we found the existence of this protein in *G. max*, *Z. mays*, *A. thaliana*, and other plants. This shows that CBL proteins are widely distributed in various higher plants, and according to their distribution in different subfamilies, we can infer that CBL proteins play an important role in the evolution of higher plants. Here, a comprehensive analysis of the soybean genome led to the identification of 15 putative *CBL* genes (Table 1). These genes were categorized into five phylogenetic branches, suggesting some degree of functional divergence among these related family members over the course of *G. max’s* evolution (Figure 1A). Comparative analyses of nine species revealed that the numbers of *CBL* genes varied significantly across the species, with the *GmCBL* gene family being the largest *CBL* gene family among the analyzed leguminous plant species (Appendix A). The sizes of the encoded GmCBL proteins were highly conserved, ranging from 24.4 to 30.65 kDa. This aligns well with what has been reported in *A. thaliana*, with AtCBLs ranging from 24.41 to 29.35 kDa in size (Table 1 and Appendix A). The 15 *GmCBLs* exhibited only limited variations in terms of nucleotide sequence length, isoelectric point, and molecular weight, although they did vary in terms of their localization to particular intracellular compartments, consistent with their potential differential functions (Table 1). Through the study of the gene structure, we found that an intron structure widely exists in *GmCBL* genes, and different *GmCBL* genes contain different intron regions (Figure 1B), which may affect the function and expression of genes, thus affecting plant growth and development and environmental stress. Multiple sequence alignment results indicated that 11 of these GmCBLs harbor C-terminal FPSF motifs (Figure 4A). Of note is the fact that the serine residues present in these FPSF motifs can be phosphorylated by CIPKs [16], supporting the potential existence of a CBL-CIPK signaling network in soybean tissues. Soybean plant tolerance has been demonstrated to be strongly dependent on N-myristoylation [17]. Intriguingly, potential N-terminal myristoylated MGCXXSK/T motifs were detected in GmCBL1, GmCBL2, GmCBL4, GmCBL11, and GmCBL13 in this study, consistent with their potential ability to regulate salt tolerance (Figure 4A). The high degree of motif conservation among these GmCBLs further emphasizes their close phylogenetic relationships [50]. All 15 *GmCBL* genes exhibit conserved motifs 1, 2, 3, 4, and 6, suggesting a potential significant biological function within the protein family. Despite evolutionary divergence among family members, these five motifs remain highly conserved, facilitating similar biological functions across various tissues for the 15 *GmCBL* genes. (Figure 2A). However, additional research will be essential to clarify the molecular functions of these particular motifs. In contrast, motif 10 was only found to be present in GmCBL5 and GmCBL8, suggesting that it may exert a more unique biological function (Figure 2A). Future studies aimed at elucidating the specific contributions of these motifs to the *GmCBL* gene family functional diversity are thus warranted.

The *CBL* gene family has been demonstrated to be characterized by duplication events in many species. Here, 13 such instances of fragment duplication were observed in the *GmCBL* family (Figure 1C), as compared to 7 fragment duplication events and one tandem repeat event reported in pecan [51]. However, this absence of any apparent tandem repeat events in the *GmCBL* family is consistent with the *CBL* gene families reported in pepper [52] and cotton plants [53]. This suggests that fragmental duplication, rather than tandem duplication, is likely the main driver of *GmCBL* gene expansion. Analyses of interspecific collinearity highlighted pronounced patterns, with the greatest collinearity being detected between the *G. soja GsCBL* and *GmCBL* gene families, followed by the *V. unguiculata VuCBL* gene family (Figure 5). The lowest levels of collinearity were observed when assessing the *Z. mays* and *O. sativa CBL* gene families, highlighting the evolutionary divergence of these families of related orthologous genes across species. The higher levels of collinearity between *GmCBLs* and *CBLs* encoded by other leguminous plants provide further confirmation of their close phylogenetic relationships. Strikingly, *GmCBL2* and *GmCBL11* exhibited collinearity with *S. bicolor*, *A. thaliana*, *O. sativa*, *Z. mays*, *P. vulgaris*, *V. unguiculata*, and *G. soja* (Appendix A), suggesting the potential emergence of these genes prior to the evolutionary divergence of these species and underscoring their conservation.

The structures and functions of plant gene promoters are inextricably linked with one another [54]. The promoters of *GmCBLs* were found to harbor a wide range of *cis*-acting elements, emphasizing the ability of these genes to respond to diverse internal and external factors (Figure 3). Of note is the fact that these genes were found to harbor many regulatory elements associated with responses to hormones and abiotic stressors, supporting the ability of *GmCBLs* to be subject to hormone-mediated regulation and to play an active role in abiotic stress responses, as has been reported previously in other species [55,56]. For instance, hormone- and abiotic-stress-responsive elements have been detected upstream of members of the *MtCBL* and *MsCBL* gene families, including LTR, ABA, MBS, and ARE elements [57]. The high numbers of elements associated with responses to ABA, SA, GA, MeJA, cold, and drought conditions upstream of *GmCBLs* suggests the potentially central role that these genes may play as coordinators of hormone signaling and stress responses. *GmCBL1* was identified as a target of particular interest, harboring several GA, SA, ABA, MeJA, and stress-response-associated elements. *GmCBL1* exhibits homology to SOS3, which is a reported enhancer of plant salt tolerance [58], further bolstering the likelihood that it can respond to a range of hormones and stressors. Meristem expression elements were also found to be enriched upstream of the *GmCBL1*, *GmCBL3*, *GmCBL5*, *GmCBL11*, and *GmCBL14* genes, which were expressed at high levels in mature tissues, indicating that the proteins they may encode may play important functional roles in the plant meristem compartment. This is consistent with interactions between *CBL4* and *CIPK25* reported previously in the root meristem compartment [59]. Changes in the expression profiles of particular genes over the course of plant growth can yield insight into their functional roles. Interactions between CBL9 and CIPK in *A. thaliana,* for example, have been found to regulate ABA responses during seed germination [60]. Maize *ZmCBLs* other than *ZmCBL6* have also been identified as regulators of seed germination and early seedling growth [61]. Here, diverse patterns of *GmCBL* gene expression were observed across tissues, emphasizing the functionally distinct roles that they play in soybean plants (Figure 4B). High *GmCBL3* and *GmCBL11* expression levels were only detected in flowers, indicating that they may help to control flowering time. *GmCBL10* and *GmCBL7,* in contrast, were specifically highly expressed in seeds, such that they may act as key regulators of seed development. As four *GmCBL* genes were expressed primarily in roots, they may act to govern root development and signal transduction activity. Similar to the interaction of CBL and CIPK25 in *A. thaliana*, these *GmCBLs* likely act in part to control auxin and cytokinin signaling in the context of root meristem development [59]. 

Recently, a wealth of evidence has emphasized the important role that *CBL* genes play as regulators of plant abiotic stress responses [62]. For instance, overexpressing *CBL4* in pigeon pea has been linked to enhanced salt stress, drought stress, and abnormal temperature resistance [63], while the *NtCBL* gene in *Nicotiana tabacum* has been established as a regulator of drought and salt stress responses [64]. Accordingly, the responsivity of *GmCBL* expression to various abiotic stressors was probed in greater detail. The differential patterns of expression for these 15 *GmCBLs* in response to salt, drought, and alkaline stress conditions offered support for the roles that these genes play in response to abiotic stressors (Figure 6). Most of these *GmCBLs* were upregulated in response to salt and alkaline stress, although *GmCBL2*, *GmCBL5*, and *GmCBL15* were significantly downregulated under these conditions, such that they may serve as negative regulators of these stress responses. As the promoters upstream of these genes harbor multiple stress- and hormone-responsive *cis*-acting elements, and as many of these genes were upregulated in response to salt and alkaline stress, this suggested that they function as active mediators of resistance to these stressors. Notably, *GmCBL1* was rapidly upregulated, beginning within 1 h of exposure to saline–alkaline conditions, suggesting it may act as a particularly robust regulator under these conditions conducive to greater stress tolerance. CBL family members have previously been shown to be highly responsive to myriad hormone signals [18]. Consistently, ABA treatment was associated with a marked upregulation of several of these *GmCBLs* (Figure 7). The vacuolar-membrane-localized *AtCBL2*, which exhibits characteristic S-acylation, has been established as an essential regulator of plant cell responses to ABA [18]. *GmCBL1* was selected as a representative member of this gene family for further studies aimed at elucidating its intracellular functions under saline–alkaline stress conditions. Overexpressing *GmCBL1* markedly enhanced the tolerance of transgenic soybean plants for saline–alkaline stress conditions, as confirmed by increases in root length and root weight (Figure 9). Ca^2+^ ion signaling within cells has been established as an important regulator of cellular redox status [65]. *GmCBL1*-OHR plants exhibited significant increases in antioxidant- and redox-related genes, similarly supporting a role for *GmCBL1* as a regulator of redox homeostasis. The Na^+^/H^+^ transporter complex has also been shown to shape the ability of plants to tolerate saline conditions [66]. Significantly increased *GmCPA1* gene expression was observed in the transgenic plants overexpressing *GmCBL1,* suggesting that this CBL family member may augment tolerance to saline–alkaline stress, at least in part by controlling Na^+^ accumulation within soybean cells. When soybean is faced with high salt stress, it responds to the challenge by calcium ion signal transmission (Figure 10). GmCBL1 protein plays an important role in this process. It can specifically recognize calcium signals and form a protein complex with CIPK. This complex plays a role in the underground part of soybean and regulates the expression of related proteins. The mechanism of this complex includes two aspects: First, it regulates the clearance of ROS by enhancing the expression of antioxidant enzymes, thereby alleviating salt-induced oxidative stress, reducing cell oxidative damage, and maintaining ROS homeostasis (Figure 9F–I). Secondly, it selectively enhances the expression of Na^+^/H^+^ transporters to regulate the homeostasis of Na^+^ ions in cells (Figure 9E). In summary, these *GmCBL* genes are inducible in response to both hormones and abiotic stress conditions, and *GmCBL1* exhibits a potent and rapid response to saline and alkaline stress conditions.

## 5. Conclusions

In conclusion, these analyses led to the identification of 15 putative *CBL* genes encoded in the soybean genome that were grouped into five phylogenetic subfamilies. The observed variability in the patterns of expression of these *GmCBL* genes, particularly under abiotic stress and hormone treatment conditions, emphasizes the specific and varied regulatory roles of these factors. *GmCBL1* was found to exhibit particularly robust transcriptional responses to saline and alkaline stress conditions, suggesting that it may function as an important regulator of saline–alkaline stress tolerance. These results enrich the current understanding of the soybean *CBL* gene family, yielding new insights into the functional diversity of these proteins with potential implications for their roles as mediators of soybean plant stress tolerance.

## Figures and Tables

**Figure 1 plants-13-01304-f001:**
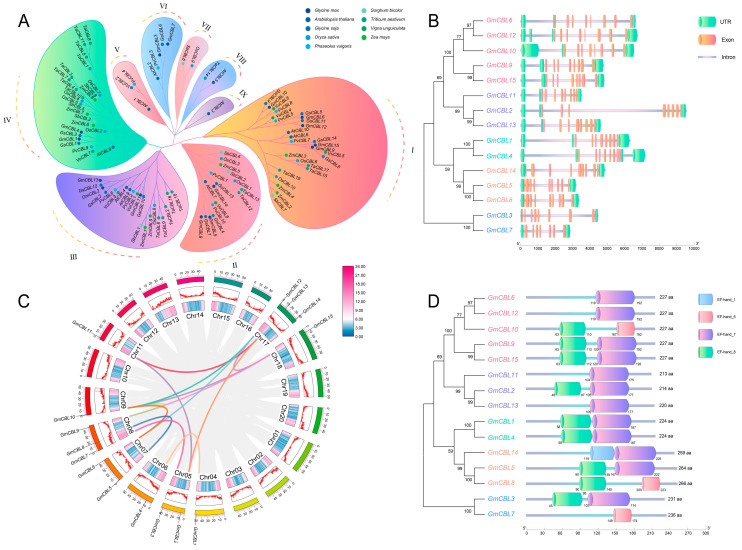
Phylogenetic, syntenic, exon–intron structure, and protein domain analyses of the *GmCBL* gene family. (**A**) CBL proteins from *G. max*, *A. thaliana*, *G. soja*, *T. aestivum*, *V. unguiculata*, *Z. mays*, *O. sativa*, *P. vulgaris*, and *S. bicolor* were used to establish a phylogenetic tree, with different colors corresponding to individual species, highlighting a range of taxonomic characteristics. (**B**) *GmCBL* gene family exon–intron organization, revealing *GmCBL* gene clusters with similar exon–intron structures. Green boxes, orange boxes, and pink lines are used to represent UTRs, exons, and introns, respectively. (**C**) Analyses of *GmCBL* gene family collinearity, with gray lines denoting collinear genomic blocks, while colored lines denote repetitive gene pair fragments. Color gradients extending from pink (high-density) to blue (low-density) represent log_2_ expression values, and chromosomes are represented by colored boxes. (**D**) Conserved domain analyses of *G. max* proteins, with blue, pink, purple, and green boxes, respectively, corresponding to EF-hand_1, EF-hand_5, EF-hand_7, and EF-hand_8 domains, all of which afford a high degree of structural diversity to these GmCBL proteins.

**Figure 2 plants-13-01304-f002:**
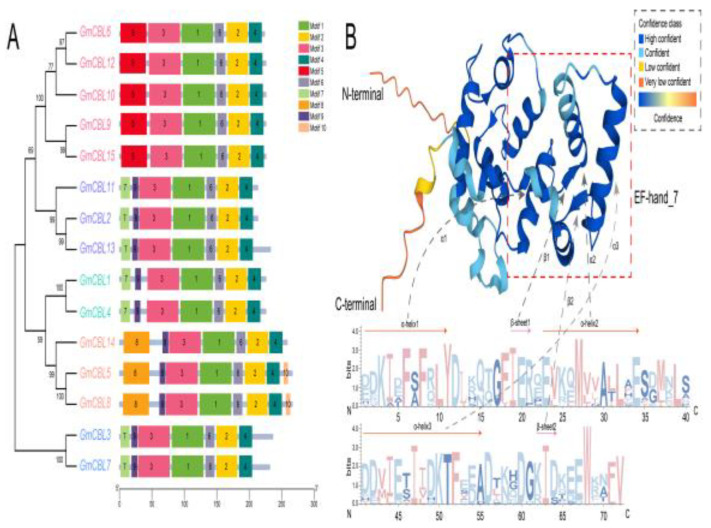
Conserved motif analyses of 3D structural models of GmCBL proteins. (**A**) Conserved motifs present within members of the *GmCBL* family are highlighted using numbered and colored boxes representing distinct motifs. (**B**) Three-dimensional structural models of the EF−hand_7 domain complexed with double-helical DNA generated with AlphaFold2 and a corresponding EF−hand_7 sequence logo generated using WebLogo.

**Figure 3 plants-13-01304-f003:**
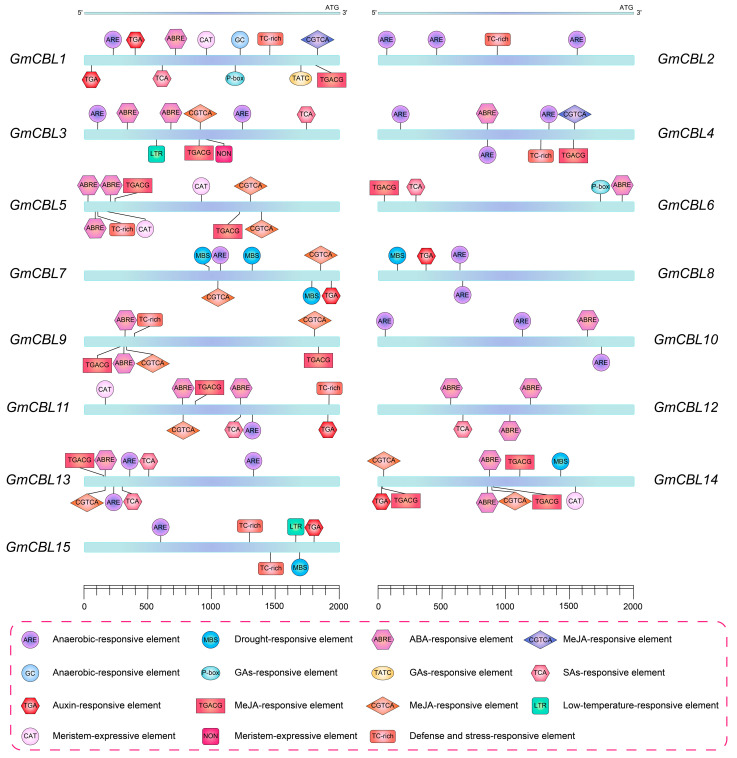
*Cis*-acting element analysis of *GmCBLs*. Predicted *cis*-acting elements were identified within the 2.0 kb promoter region upstream from the start codon for each of the identified *GmCBL* genes, with boxes of different colors being used to represent the relative positions of these elements.

**Figure 4 plants-13-01304-f004:**
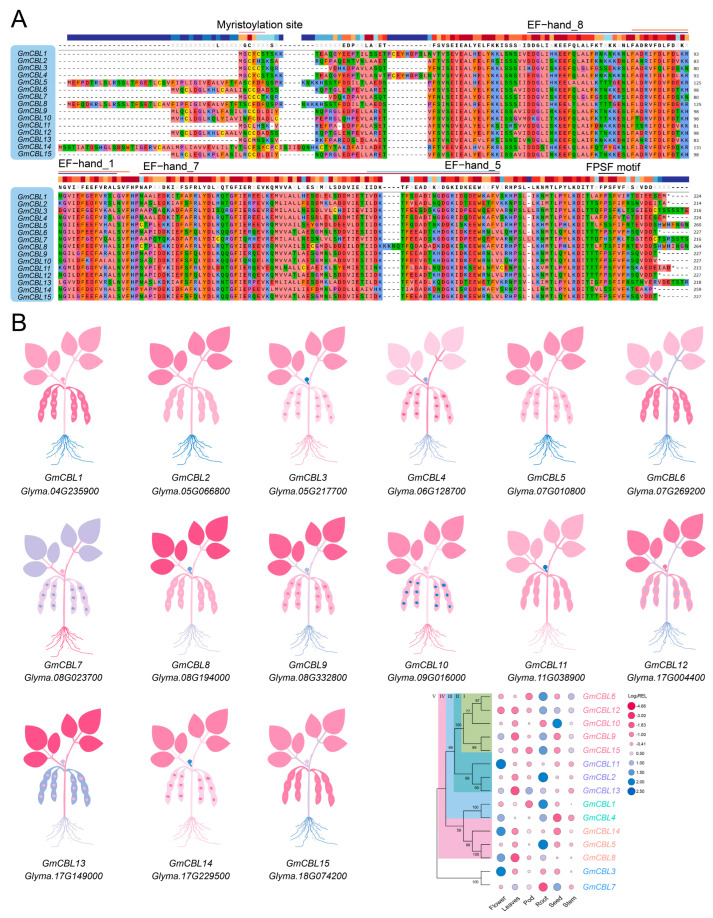
*GmCBL* domain alignment and expression pattern analyses. (**A**) *GmCBL* gene conserved domains, including the EF−hand_1, EF−hand_5, EF−hand_7, and EF−hand_8 domains, were identified through alignment. Pink and black lines, respectively, indicate the myristoylation site and the FPSF motif. (**B**) *GmCBL* gene expression patterns were analyzed in a range of soybean tissues in the Phytozome database, with TBtools having been used to establish an expression heatmap. The color scale represents log_2_ expression levels, with pink and blue, respectively, corresponding to high- and low-abundance transcripts.

**Figure 5 plants-13-01304-f005:**
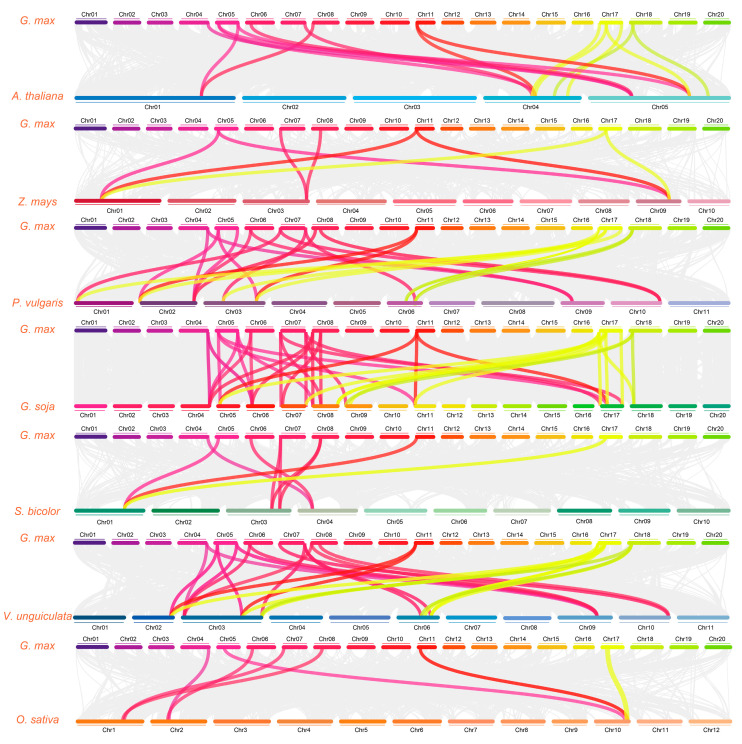
Synteny analyses *GmCBL* genes and *CBL* genes from seven other plant species. Gray background lines denote collinear genomic blocks within the genomes of *G. max* and other plant species, while different colors indicate syntenic *GmCBL* gene pairs.

**Figure 6 plants-13-01304-f006:**
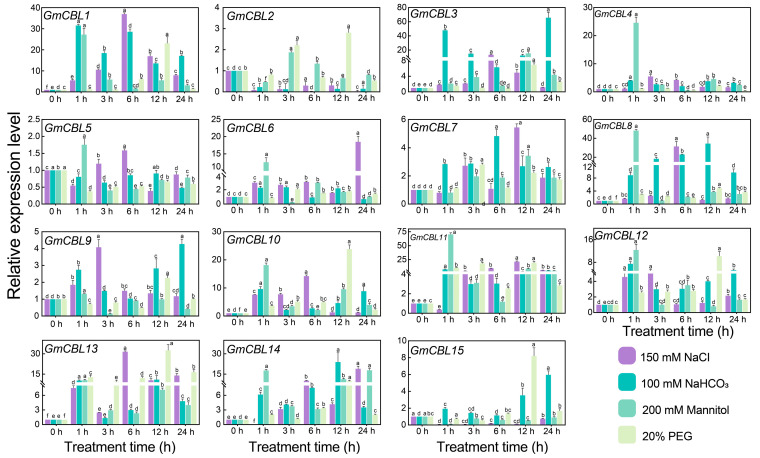
*GmCBL* expression profiling under abiotic stress conditions. *GmCBL* expression patterns were assessed in the roots of soybean plants treated with 150 mmol L^−1^ NaCl, 200 mmol L^−1^ mannitol, 20% PEG,100 mmol L^−1^ NaHCO_3_, or water for 0, 1, 3, 6, 12, or 24 h, analyzing the expression of these genes in the absence of stress as a reference control. The 2^−∆∆CT^ method was employed for relative quantification, analyzing three replicates per tissue. Different letters represent significant differences (*p* < 0.05, Duncan’s test).

**Figure 7 plants-13-01304-f007:**
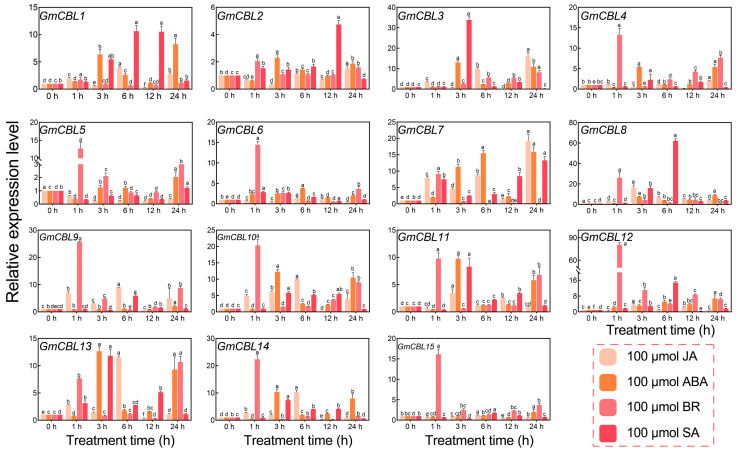
*GmCBL* expression profiling in response to hormone stimulation. The patterns of *GmCBL* gene expression in the roots of soybean plants that had been exposed to SA, BR, ABA, or JA (100 μmol L^−1^) or water as a control treatment were assessed at 0, 1, 3, 6, 12, or 24 h post stimulation, analyzing the expression of these genes in the absence of stress as a reference control. The 2^−∆∆CT^ method was employed for relative quantification, analyzing three replicates per tissue. Different letters indicate significant differences between treatments at the *p* < 0.05 level (Duncan’s test).

**Figure 8 plants-13-01304-f008:**
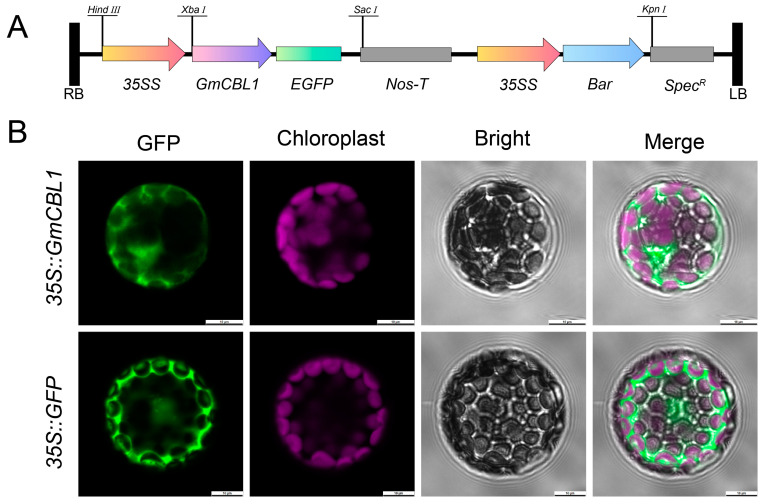
Subcellular localization analysis of transient expression of *GmCBL1*-GFP fusion protein in *A. thaliana* protoplast. (**A**) A schematic overview of the utilized pSOY1-*GmCBL1* vector, with arrows indicating expression elements and horizontal lines representing vector restriction sites. Orange represents promoter, purple represents insertion gene, green represents fluorescent protein, blue represents *bar* gene, gray represents antibiotics, and black represents terminator. (**B**) The subcellular localization of *GmCBL1* in *A. thaliana* protoplasts was analyzed, and the confocal image represents the subcellular localization of GFP and *GmCBL1*-GFP. Scale bar = 10 μm.

**Figure 9 plants-13-01304-f009:**
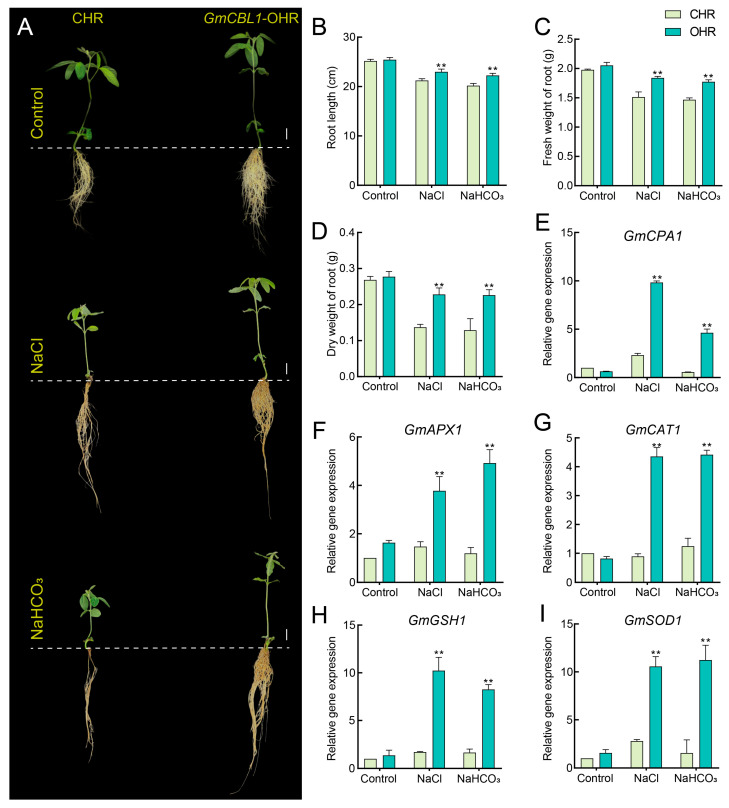
Overexpressing *GmCBL1* increases the resistance of transgenic soybean plants to saline–alkaline conditions. (**A**) Soybean plants with *GmCBL1*-overexpressing hairy roots (OHR) or control hairy roots (CHR) were treated with NaCl (0 or 75 mM) and NaHCO_3_ (75 mM) for 5 days, and their performance was evaluated. *Agrobacterium rhizogenes* was used to establish OHR plants, and over 10 hairy root lines were analyzed. (**B**) Root length. (**C**) Fresh weight of root. (**D**) Dry weight of root. (**E**–**I**) Expression of redox-related genes (*GmAPX1*, *GmCAT1*, *GmGSH1*, and *GmSOD1*) and Na^+^/H^+^ antiporter gene (*GmCPA1*) in OHR and CHR plants was assessed 12 h after exposure to salt and alkali stress, analyzing three biological replicates per tissue. ** *p* < 0.01 vs. CHR plants; Student’s *t*-test. Data are means ± SD.

**Figure 10 plants-13-01304-f010:**
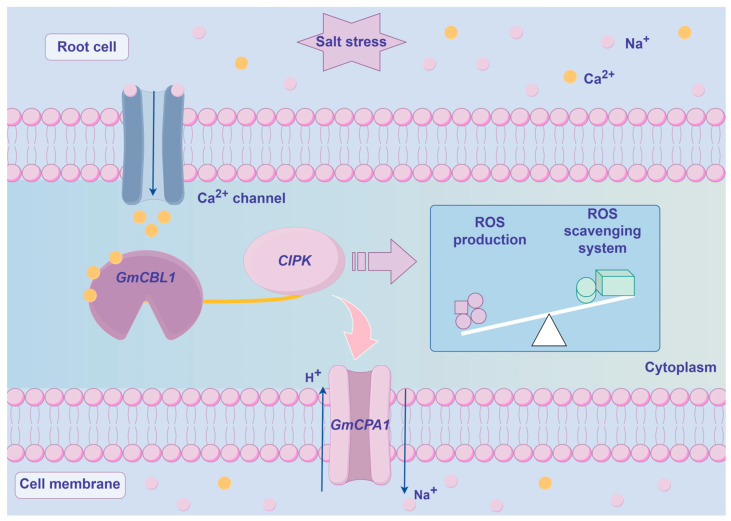
The model of *GmCBL1* gene involved in the regulation of ROS level under saline–alkaline stress. When soybean root cells are exposed to saline–alkaline stress, calcium channel proteins situated on the cell membrane are activated, leading to the influx of calcium ions into the cytoplasm. This influx results in an elevation of calcium concentration within the cytoplasm and subsequently initiates calcium signal transduction pathways. The calcium ions interact with the GmCBL1 protein, leading to the activation of the CBL-CIPK signal pathway, which in turn triggers the antioxidant defense system. Soybean plants are able to mitigate the effects of saline–alkaline stress by upregulating the expression of antioxidant enzymes, thereby reducing oxidative damage. Arrows are employed to denote the direction of change.

**Table 1 plants-13-01304-t001:** Basic characteristics of the 15 identified soybean *CBL* genes.

Gene Name	Gene ID	Gene Location	ORF Length (bp)	Protein Length	Isoelectric Point	Molecular Weight (kDa)	Subcellular Localization
*GmCBL1*	*Glyma.04G235900*	Chr4 50399334…50405613	672	224	4.4	25.68	Cytoplasmic
*GmCBL* *2*	*Glyma.05G066800*	Chr5 6731396…6740875	642	214	4.49	24.22	Nuclear
*GmCBL* *3*	*Glyma.05G217700*	Chr5 39770219…39774659	693	231	4.59	26.1	Nuclear
*GmCBL* *4*	*Glyma.06G128700*	Chr6 10581579…10588773	672	224	4.42	25.55	Cytoplasmic
*GmCBL* *5*	*Glyma.07G010800*	Chr7 824696…827823	792	264	4.76	30.44	Cytoplasmic
*GmCBL* *6*	*Glyma.07G269200*	Chr7 44230392…44236986	681	227	4.39	25.78	Extracellular matrix
*GmCBL* *7*	*Glyma.08G023700*	Chr8 1884590…1887532	708	236	4.53	26.73	Nuclear
*GmCBL* *8*	*Glyma.08G194000*	Chr8 15628395…15631841	798	266	4.76	30.65	Cytoplasmic
*GmCBL* *9*	*Glyma.08G332800*	Chr8 44992808…44997458	681	227	4.45	26.04	Cytoplasmic
*GmCBL* *10*	*Glyma.09G016000*	Chr9 1233158…1239867	681	227	4.64	25.93	Nuclear
*GmCBL* *11*	*Glyma.11G038900*	Chr11 2773817…2777295	639	213	4.31	24.40	Nuclear
*GmCBL* *12*	*Glyma.17G004400*	Chr17 434853…441609	681	227	4.40	25.80	Cytoplasmic
*GmCBL* *13*	*Glyma.17G149000*	Chr17 12354772…12359193	660	220	4.72	25.24	Cytoplasmic
*GmCBL14*	*Glyma.17G229500*	Chr17 38443770…38448625	777	259	4.44	29.26	Chloroplast
*GmCBL15*	*Glyma.18G074200*	Chr18 7033194…7037977	681	227	4.50	26.07	Cytoplasmic

## Data Availability

Data are contained within the article and Appendix A.

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
