# Peer review of "Genome-Wide Identification of Members of the Soybean *CBL* Gene Family and Characterization of the Functional Role of *GmCBL1* in Responses to Saline and Alkaline Stress"

_plants, 2024, doi:10.3390/plants13101304_

Round 1

Reviewer 1 Report

Comments and Suggestions for Authors

This manuscript titled “Genome-Wide Identification of Members of the Soybean CBL Gene Family and Characterization of the Functional Role of GmCBL1 in Responses to Saline and Alkaline Stress” presents essential new data about the CBL gene family. The manuscript indicates that the CBL gene is ambiguous regarding soyabean. There are several shortcomings that should be resolved.

Some major remarks:

#1: Could you please include GmCBL1 GFP localization to demonstrate that the prediction is valid?

#2: Please create a hypothetical pathway for how GmCBL1 responds to saline and alkaline stress and describe it in the manuscript.

Some minor remarks:

Lines 41, 283: Give some references that analyzed the 2kb promoter region upstream from the start codon.

Lines 245, Figure 1: 1A, could you please add circular phylogenetic analysis with bootstrap in the node and discuss it in the manuscript? The author can also make categories of the bootstrap, like: a high bootstrap value suggests strong support for the grouping at that node, while a lower value indicates less confidence in the grouping. The authors can keep the circular bootstrap in the supplementary. 1B, Intron Phase, should be analyzed and described in the manuscript.

Lines 275, Figure 2A: Could you please make a representable number inside the motif box and the top few motif (5–6) in the manuscript and explain to them why it’s important?

Author Response

1. Could you please include GmCBL1 GFP localization to demonstrate that the prediction is valid?

Thank you very much for your help on pointing out our problems. We think your proposal is very good. We supplemented the subcellular localization experiments of GmCBL1 in Arabidopsis protoplasts to validate the prediction of GmCBL1. The content was supplemented with materials and methods and results, was yellowed.

(Materials and Methods, Page 4, lines 174; Materials and Methods, Page 4, lines 185-187; Results, Page 16, lines 416-428.)

2. Please create a hypothetical pathway for how GmCBL1 responds to saline and alkaline stress and describe it in the manuscript.

Thank you very much for your help on pointing out our problems. Your advice is very good. I have added the corresponding content in this revised draft and emphasized it in yellow.

(Discussion, page 21, lines 590-600.)

3. Give some references that analyzed the 2kb promoter region upstream from the start codon.

Once again, we thank the reviewer for this valuable suggestion. Due to our carelessness, we forgot to add references to analyze the 2kb promoter region upstream of the initial codon in the material method. We added several references in the revised draft to make the results of the article more reliable.

(Materials and Methods, Page 3, lines 153.)

4. Figure 1: 1A, could you please add circular phylogenetic analysis with bootstrap in the node and discuss it in the manuscript? The author can also make categories of the bootstrap, like: a high bootstrap value suggests strong support for the grouping at that node, while a lower value indicates less confidence in the grouping.

Thank you for your nice suggestions on improving our manuscript. We have supplemented the relevant contents in the latest revised draft and marked them yellow, and carefully revised the relevant contents.

(Supplementary material S1; Results, Page5, lines 200; Discussion, Page18, lines 454-458.)

5. 1B, Intron Phase, should be analyzed and described in the manuscript.

Dear Editor, thank you very much for your help on pointing out our problems. We have added the relevant content and marked it yellow in the latest revised draft, and carefully revised the relevant content.

(Discussion, Page 18, lines 470-473)

6. Could you please make a representable number inside the motif box and the top few motif (5–6) in the manuscript and explain to them why it’s important?

Thank you very much for your help on pointing out this problem. We agree with you very much, and we have added relevant contents. In the revised draft, the corresponding contents are supplemented and marked yellow.

(Figure 2, Discussion, Page 18, lines 470-473.)

Reviewer 2 Report

Comments and Suggestions for Authors

Thank you for the manuscript!

Please provide an explanation of the experimental design used, specifying whether it was conducted in a field or greenhouse and the latitudes and longitudes in the experiment site. As your sample includes multiple levels of studied factors including stress conditions, hormones, and time points, please use ANOVA or longitudinal analysis to determine if there is an interaction effect between stress conditions and hormones. If such an interaction effect is statistically significant, please discuss it instead of the main effects.

It is essential to avoid using multiple t-tests in the same experiment because it can increase the likelihood of making Type I errors that exceed the acceptable limit of 5%. Instead, it is recommended to use ANOVA analysis, which helps to control these errors and maintain the Type I error rate at 5%. Therefore, I suggest you reanalyze the stress conditions and hormone experiments using ANOVA analysis to obtain more reliable results.

It is important to use the full scientific name, including genus and species, when introducing a scientific name, such as Glycine max L., and later, a shortened version like G. max can be used.

Could you please provide readers with the explanations for Figure 8's figures B, C, and D in the description?

Comments on the Quality of English Language

I didn't notice any significant language problems.

Author Response

1. Please provide an explanation of the experimental design used, specifying whether it was conducted in a field or greenhouse and the latitudes and longitudes in the experiment site.

Dear Editor, thank you very much for your help on pointing out our problems. We have carefully supplemented the relevant contents in the latest revised draft.

(Materials and Methods, Page 3, lines 122-123.)

2. As your sample includes multiple levels of studied factors including stress conditions, hormones, and time points, please use ANOVA or longitudinal analysis to determine if there is an interaction effect between stress conditions and hormones. If such an interaction effect is statistically significant, please discuss it instead of the main effects.

Thank you very much for your help on pointing out our problems. We agree with you very much, and we have added relevant contents. We have carefully supplemented the relevant contents in the latest revised draft.

(Figure S4; Results, Page15, lines 404-407.)

3. It is essential to avoid using multiple t-tests in the same experiment because it can increase the likelihood of making Type I errors that exceed the acceptable limit of 5%. Instead, it is recommended to use ANOVA analysis, which helps to control these errors and maintain the Type I error rate at 5%. Therefore, I suggest you reanalyze the stress conditions and hormone experiments using ANOVA analysis to obtain more reliable results.

Dear Editor, thank you very much for your help on pointing out our problems. We are really sorry for using the wrong method for analysis due to our negligence, and the relevant contents have been added and modified according to your opinions in the latest revised draft.

(Figure 6 and Figure 7)

4. It is important to use the full scientific name, including genus and species, when introducing a scientific name, such as Glycine max L., and later, a shortened version like G. max can be used.

Dear Editor, thank you very much for your help on pointing out our problems. We have revised the relevant contents in the latest revised draft.

(Key words, Page1, lines 21.)

5. Could you please provide readers with the explanations for Figure 8's figures B, C, and D in the description?

Thank you very much for your time on reviewing our manuscript. We have carefully read the nice comments from you and found that these suggestions are very helpful for us to improve our manuscript. We have revised the relevant contents in the latest revised draft.

(Discussion, Page 20 lines 578-586.)

Round 2

Reviewer 1 Report

Comments and Suggestions for Authors

The question has been well addressed in the author's response.
It is quite excellent that the writers conduct extra experiments.
Thanks.